# Covalently linked dengue virus envelope glycoprotein dimers reduce exposure of the immunodominant fusion loop epitope

Alexander Rouvinski[1,2,*,†], Wanwisa Dejnirattisai[3,*], Pablo Guardado-Calvo[1,2,*], Marie-Christine Vaney[1,2], Arvind Sharma[1,2], Stéphane Duquerroy[1,2,4], Piyada Supasa[3], Wiyada Wongwiwat[3], Ahmed Haouz[5], Giovanna Barba-Spaeth[1,2], Juthathip Mongkolsapaya[3,6], Félix A. Rey[1,2] & Gavin R. Screaton[3]

A problem in the search for an efficient vaccine against dengue virus is the immuno-dominance of the fusion loop epitope (FLE), a segment of the envelope protein E that is buried at the interface of the E dimers coating mature viral particles. Anti-FLE antibodies are broadly cross-reactive but poorly neutralizing, displaying a strong infection enhancing potential. FLE exposure takes place via dynamic 'breathing' of E dimers at the virion surface. In contrast, antibodies targeting the E dimer epitope (EDE), readily exposed at the E dimer interface over the region of the conserved fusion loop, are very potent and broadly neutralizing. We here engineer E dimers locked by inter-subunit disulfide bonds, and show by X-ray crystallography and by binding to a panel of human antibodies that these engineered dimers do not expose the FLE, while retaining the EDE exposure. These locked dimers are strong immunogen candidates for a next-generation vaccine.

[1] Institut Pasteur, Département de Virologie, Unité de Virologie Structurale, 75724 Paris Cedex 15, France. [2] CNRS UMR 3569 Virologie, 75724 Paris Cedex 15, France. [3] Division of Immunology and Inflammation, Department of Medicine, Hammersmith Hospital Campus, Imperial College London, w12 0NN London, UK. [4] Université Paris-Sud, Faculté des Sciences, 91405 Orsay, France. [5] Institut Pasteur, Protéopôle, CNRS UMR 3528, 75724 Paris Cedex 15, France. [6] Dengue Haemorrhagic Fever Research Unit, Office for Research and Development, Faculty of Medicine, Siriraj Hospital, Mahidol University, Bangkok 10700, Thailand. * These authors contributed equally to this work. † Present address: Department of Microbiology and Molecular Genetics, Kuvin Center for the Study of Infectious and Tropical Diseases, Institute for Medical Research Israel-Canada, The Hebrew University-Hadassah Medical School, The Hebrew University of Jerusalem, Jerusalem, Israel. Correspondence and requests for materials should be addressed to J.M. (email: j.mongkolsapaya@imperial.ac.uk) or to F.A.R. (email: felix.rey@pasteur.fr) or to G.R.S. (email: g.screaton@imperial.ac.uk).

Certain members of the Flavivirus genus are the most important arthropod borne viral pathogens of humans, causing increasingly serious disease outbreaks. The flaviviral disease that imposes the highest toll on society is dengue, which is caused by four different viruses termed serotypes DENV1-4, differing by 30–35% in amino acid sequence of their envelope proteins[1]. It is estimated that the annual global incidence is 390 million cases, of which 96 million are clinically apparent, with around 25 thousand deaths[2]. Several factors drive the pandemic, including globalization, the spread of the Aedes mosquito vector around the world, inadequately planned urbanization and the absence until recently of a licensed vaccine or anti-DENV therapeutics[3–5]. Zika virus (ZIKV) is also transmitted by Aedes mosquitos, and among the flaviviruses, its envelope protein is closest in amino acid sequence (about 56% identity) to that of the DENVs than to other flaviviruses[6].

The hallmark of severe DENV infection is increased capillary permeability, causing plasma leakage and bleeding, leading to haemodynamic compromise and DENV shock syndrome[1]. Untreated, severe disease can lead to a mortality of up to 20%, but with expert management, primarily fluid replacement, it can be reduced to below 1%. DENV has caused explosive epidemics, putting huge stress on healthcare systems in endemic countries. Although several DENV control strategies are being evaluated, it is generally agreed that an effective vaccine available to all age groups is required to make serious inroads into the burden of disease[1,5].

Infection with one serotype of DENV results in the generation of lifelong immunity to reinfection with that serotype, but not to the others[1]. As all four DENV serotypes frequently co-circulate, or cyclically replace each other, multiple infections are the norm in endemic countries. Well-controlled epidemiological studies demonstrate that most severe DENV infections occur in individuals who are experiencing a secondary or sequential DENV infection[7].

The hypothesis of antibody dependent enhancement (ADE) posits that pre-existing heterologous antibodies generated during a primary infection may not efficiently neutralize a secondarily encountered virus[8]. Instead, the virus may be opsonized and targeted for uptake into Fc-receptor bearing cells such as monocytes and macrophages, which are important sites of DENV replication in vivo, and therefore lead to an increase in viral production.

There have now been a number of descriptions of human monoclonal antibodies in DENV infection[9–16]. The immunodominant epitope to DENV appears to be the FLE, a linear epitope spanning the fusion loop, which is highly conserved in flaviviruses. Anti-FLE mAbs are frequently cross-reactive across all DENV serotypes and also across flaviviruses[9,12,17]. Because the FLE is sub-optimally presented by mature flaviviruses, anti-FLE mAbs often show poor neutralization but potently induce ADE[9,12,17–19]. PrM-specific antibodies are also a major component of the memory B cell response to DENV; these antibodies show poor neutralization (maximum 30–50%) even at high concentration[20].

We have recently described the cloning of a large panel of anti-E mAbs from DENV infected patients[12]. One third of the antibodies generated recognize a conformational, quaternary epitope and bind poorly to the recombinant E ectodomain (termed sE), which is mainly monomeric in solution. Many of these antibodies showed broad and potent neutralization of all four DENV serotypes, being among the most potent described to date. Structural characterization has shown that these mAbs bind to the E dimers at the virion surface, to a site that we termed the E-dimer epitope (EDE). In addition, we have recently discovered that the epitope recognized by some anti-EDE antibodies is also conserved in the ZIKV E-dimer, leading to equally potent neutralization, making the EDE also a potential target for ZIKV vaccines[6,21]. The observation that broadly neutralizing antibodies can be produced during DENV infection suggests that using subunit vaccines targeting the EDE is an alternative route for vaccines.

We describe here covalently linked dengue envelope dimers that can be produced in the absence of prM and which also reduce exposure of the immunodominant FLE aiming to suppress the generation of unwanted yet ADE promoting antibody responses.

## Results

**Anti-EDE mAbs can stabilize the E-dimer.** Binding of anti-EDE antibodies to sE requires the presence of head-to-tail-dimers that form only at high concentrations in solution—such as the concentrations used for crystallization, which mimic the very high effective E protein concentration at the surface of virus particles. This is exemplified when monomeric recombinant sE protein is plate bound in ELISA assays, where antibodies such as those to FLE bind well, while those reacting to the EDE do not (Fig. 1a). But when the ELISA is performed in reverse (capture ELISA) and anti-EDE mAb is bound to the plate it is able to capture sE protein, by shifting the monomer-dimer equilibrium (Fig. 1b), which cannot be achieved when the monomeric form is immobilized.

Anti-EDE antibodies can be divided into two subclasses depending on their sensitivity to the presence of the N-linked glycan at position 153, which is required for anti-EDE2 mAbs binding but not required for the anti-EDE1 mAbs binding[12]. In the capture ELISA assays described above, EDE1 C8 and EDE2 A11 antibodies were able to drive dimer formation at lower concentrations of E protein compared to EDE1 C10 and EDE2 B7, which probably represents a higher affinity of interaction with the recombinant sE-dimer (Fig. 1b). The ability of anti-EDE mAbs to drive dimer formation is further exemplified by size exclusion chromatography together with multi-angle static light scattering (MALS), where EDE1 C8 or EDE2 A11 Fabs assemble sE into a sE dimer resulting in the formation of sE/Fab heterotetramer (sE dimer with two bound Fab molecules) (Fig. 1c).

**FLE and EDE mAbs compete to bind DENV particles.** The X-ray structures of both anti-EDE1 and anti-EDE2 mAbs bound to DENV2 sE showed that they bind at the E dimer interface, at a site where the fusion loop—which is present at the tip of domain II—interacts with domains I and III from the opposite E subunit in the dimer. The EDE epitope thus spans residues from all three domains of E, including the fusion loop but in a conformation such that the non-polar Trp101 and Leu107 side chains are buried at the interface with domain III (ref. 22). In contrast, the FLE antibodies only require residues at the tip of domain II for binding, and specifically recognize the side chains of the fusion loop residues that are buried in the E dimer[23,24]. The anti-EDE and anti-FLE mAbs thus bind, when these side chains are buried or exposed, respectively. It is therefore important to understand the extent to which binding of these two classes of antibodies compete with each other for binding DENV particles.

We therefore devised a competition ELISA whereby DENV was captured on ELISA plates and the binding of biotinylated anti-EDE or anti-FLE mAbs assessed in the presence of non-biotinylated mAb competitor (Fig. 2a). In a first series of experiments we fixed the concentration of biotinylated antibody at 1 µg ml$^{-1}$ and added an increasing concentration of non-biotinylated antibody. These assays were performed on two

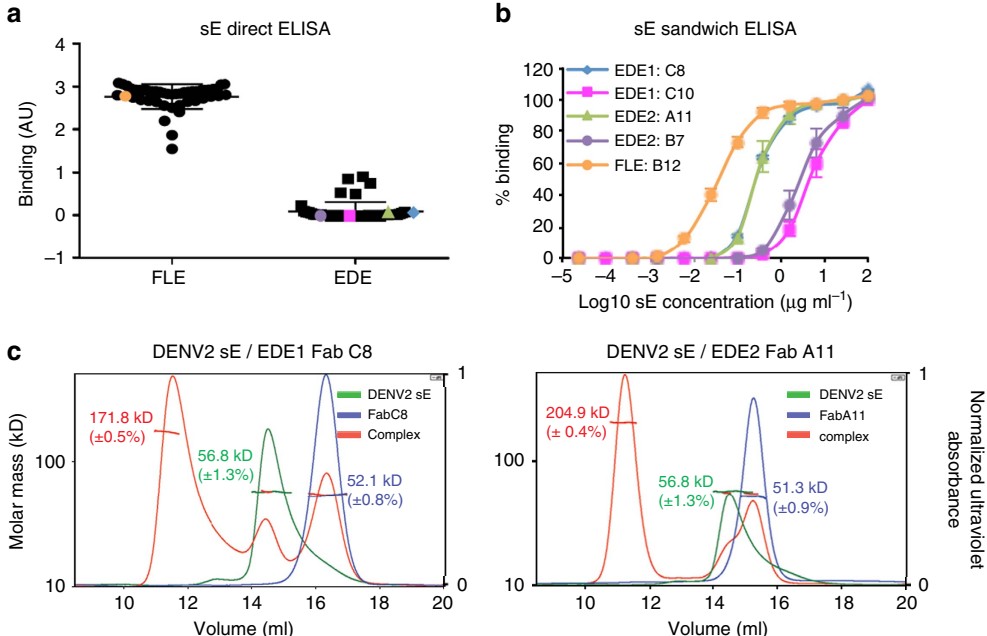

**Figure 1 | EDE mAbs can stabilize DENV envelope dimers.** (**a**) Binding of anti-FLE and anti-EDE mAbs to monomeric sE protein was determined by direct ELISA with sE coated to the ELISA plate. Results are expressed as mean of binding in arbitrary units (AU) from three independent experiments. The mAbs used in **b** are indicated by dots of the same colour. (**b**) The ability of selected mAbs to bind and assemble dimers was assessed by indirect ELISA. ELISA plates were coated with four anti-EDE mAbs and one anti-FLE mAb control, which binds monomeric sE. Plates were then incubated with a titration of soluble Strep-tagged sE monomer, bound sE was revealed using ALP-labelled StrepTactin. The data are shown as mean ± s.e.m. from three independent experiments. (**c**) SEC/MALS analysis of isolated DENV2 sE, isolated Fab fragments and DENV2 sE with anti-EDE1 Fab C8 (left panel) and anti-EDE2 Fab A11 (right panel) mAbs. The molecular weight determined by MALS is indicated, corresponding to the y axis on the left. The ultraviolet absorbance was normalized such that the highest peak is set to 1 (y axis on the right).

different virus preparations, DENV2 produced either in the mosquito cell line C6/36 or in primary human myeloid derived dendritic cells (DC). C6/36 produced viruses have a higher content of uncleaved prM compared to DENV produced in DC (Supplementary Fig. 1)[12,20]. This difference in prM leads to higher binding and neutralization of C6/36 DENV than DC-DENV by anti-FLE mAbs[12]. The neutralization of DC-DENV by anti-FLE mAbs is however incomplete with a plateau at around 80% neutralization[12]. In contrast to anti-FLE mAbs, anti-EDE mAbs are able to bind and fully neutralize DENV produced in both C6/36 cells and DC[12].

Using C6/36 cell derived DENV particles, the anti-FLE mAbs were the most effective competitors for binding to virions and were able to prevent binding of anti-EDE mAbs at high concentrations, whereas anti-EDE mAbs as expected, could compete with themselves for binding, but not with anti-FLE mAbs (Fig. 2b). When tested on low prM containing DC-DENV, anti-FLE mAb was less effective competitor for anti-EDE binding and anti-EDE mAbs were able to compete off anti-FLE binding (Fig. 2c).

In the second series of experiments we preincubated virions with biotinylated or non-biotinylated anti-FLE or anti-EDE mAbs and then looked for the ability of competitor antibody to displace bound antibody over a time course of incubation. These experiments demonstrate that once bound to DENV the interaction with both anti-FLE and anti-EDE mAbs was very stable and could not be displaced (Supplementary Fig. 2a–c).

**Stabilisation of the E-dimer.** We used a structure-based approach to identify pairs of residues at the dimer interface located such that their replacement by cysteine could potentially lead to formation of inter-subunit disulfide bonds and improve sE-dimer stability. We analysed the crystal structure of the

DENV2 sE dimer (DENV2 sE in complex with EDE2 B7 Fab, PDB accession code 4UT6 (ref. 22)) with the MODIP server (http://caps.ncbs.res.in/iws/modip.html)[25], which identified four pairs of amino acids facing each other across the dimer interface with $C_\beta$–$C_\beta$ distances under 4.5 Å. These were residues S255, A259, F108/T315 and L107/A313. S255 and A259 lying by the molecular two-fold axis, such that they face themselves in the dimer, and therefore only one mutation to cysteine is required to form a disulfide bond. The other residues are away from the molecular two-fold axis, at the interface between the fusion loop (which spans residues 98–110) and domain III, and require two mutations to cysteine and thus form two inter-monomer disulfide bonds. The MODIP server ranks the pairs from A to D depending on the adequacy of their geometry to make a disulfide bond, where A is the highest score (meaning a high probability of forming a disulfide bond in the analysed static PDB model) and D the poorest. The location of these residues in the structure of the DENV2 sE dimer, along with the respective MODIP scores, is indicated in Fig. 3a. None of the predicted disulfide had the highest score, indicating a non-optimal geometry. Nevertheless, as the DENV E protein has been shown to be flexible and has several hinge angles, it is likely that its polypeptide chain can adjust to the required geometry to make some of the disulfide bonds.

Production of these four cysteine mutants (two single and two double mutants) in *Drosophila* S2 cells showed that the best yields in disulfide linked sE dimers resulted from the A259C construct followed by the double mutant L107C/A313C (Fig. 3b), suggesting that the chain can adjust the geometry in these two cases, but not in the other two mutants. Although these mutants produced also a fraction of monomer and also of high-molecular weight aggregates, they resulted in at least 2 mg of covalent dimer produced by litre of S2 cell culture, whereas the other two mutants led essentially to only aggregates (Fig. 3b).

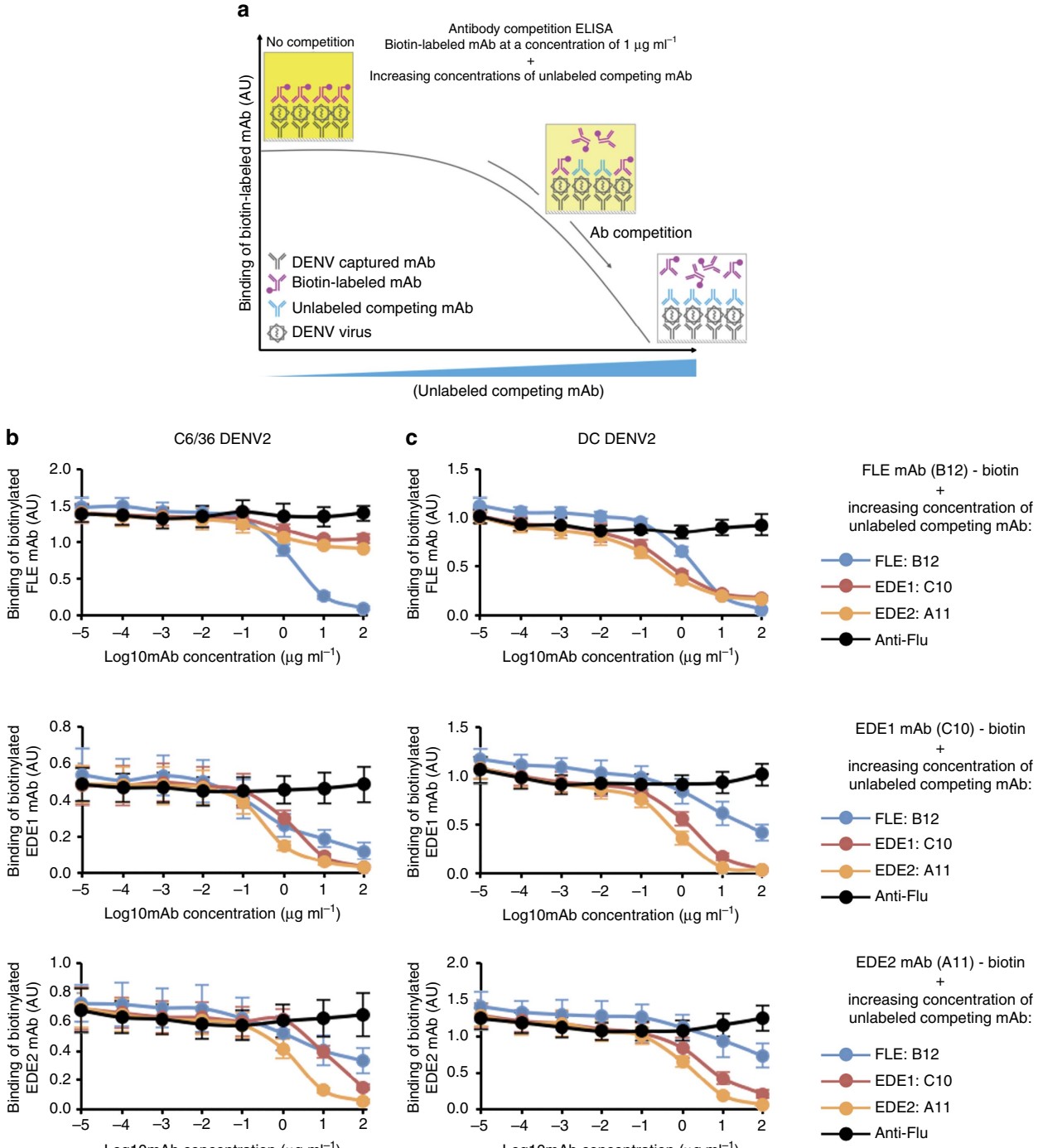

**Figure 2 | Competition between anti-FLE and anti-EDE mAbs.** (**a**) Description of the schematic procedure of antibody replacement ELISA.
(**b,c**) Competition for binding to dengue virions of anti-FLE mAb-B12, anti-EDE1 mAb-C10 and anti-EDE2 mAb A11. ELISA plates were coated with DENV2 virions produced in C6/36 cells (high prM) (b) and DC (low prM) (c) captured by murine mAb 2C8 which binds to EDIII of DENV2. Plates were then incubated with a pair of antibodies; one of which was biotinylated at a concentration of $1 \mu g \, ml^{-1}$ and a second antibody which was added in increasing concentrations. Binding of biotinylated antibody was revealed by ALP-conjugated Streptavidin. The data are shown as mean ± s.e.m. from three independent experiments.

Analysis by size exclusion chromatography (SEC) together with multi-angle static light scattering (MALS) of the A259C mutant showed that it eluted in a peak corresponding to a molecular weight of 93 kDa, as expected for a sE dimer. Under the same conditions, wild type DENV2 sE eluted as two peaks, both corresponding to a monomer (48 kDa MW) according to the molecular mass measured by MALS (Fig. 3c). The main peak

(on the right, in blue) elutes late, indicating interaction with the support, similar to what has been reported previously for other homologous class II viral fusion proteins when manipulated as monomers with the fusion loop exposed[26]. The double mutant L107C/A313C also eluted as a dimer and the peak overlapped with that of A259C (Fig. 3d), as expected. This was also confirmed by SDS–polyacrylamide gel electrophoresis

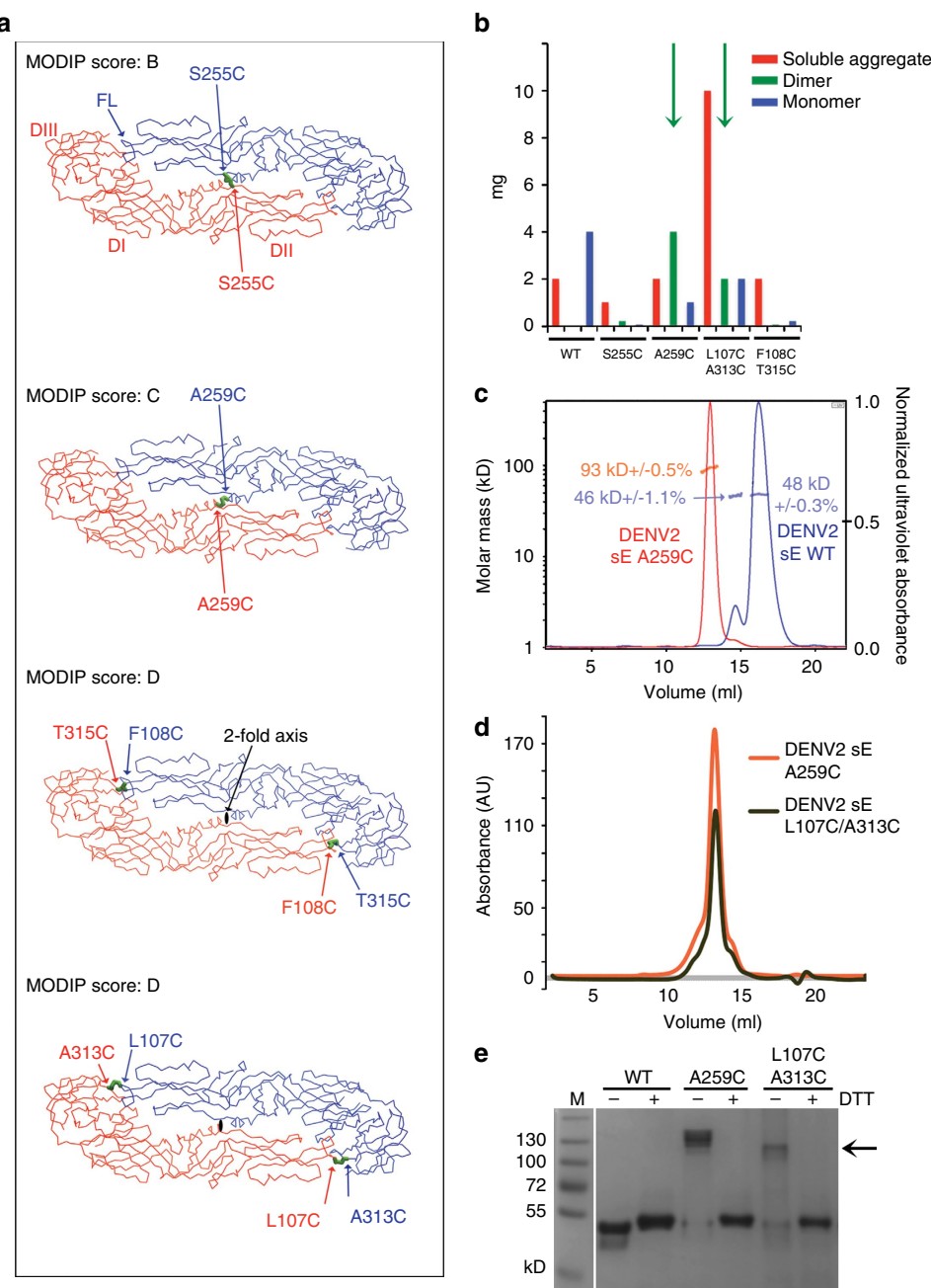

**Figure 3 | Engineering covalently linked E-dimers.** (**a**) Localization of the residues identified by MODIP susceptible to form inter-chain disulfide bonds upon mutation to cysteine. The sE dimer is coloured by subunit, with the MODIP residue pairs indicated in the corresponding colours. A disulfide bond is modelled and is shown as green sticks. The MODIP score, indicated for each residue pair, is a measure of favourability of the geometry of the selected amino acids for disulfide bond formation where A is best and D is worst. (**b**) Histogram showing the approximate yields in mg per litre of S2 cell culture of DENV2 FGA02 sE protein eluting as monomer, dimer and aggregates separated by SEC for wild type and for the four cysteine mutants presented in **a**. The yields of covalent dimers are shown in green bars (highlighted with green arrows when sufficient yields for further studies were obtained). (**c**) MALS analysis of DENV2 A259C sE (red trace). The fractions eluting as dimer in a first step of SEC (which eliminated monomers and aggregates) was re-run by SEC and then superposed to the elution profile of DENV2 WT sE (blue trace). The ultraviolet absorbance was normalized such that the highest peak of each run is set to 1 (y axis on the right). The molecular weight determined by MALS is indicated, corresponding to the y axis on the left. (**d**) SEC elution profile of L107C/A313C sE superposed to that of A259C sE, showing that the peaks are at the same elution volume, which corresponds to a dimer characterized by MALS in **b**. As in **c** the peaks corresponding to monomer and aggregates were eliminated in an initial SEC run. (**e**) Coomassie stained SDS-PAGE run of sE WT and sE mutants of DENV2 (in the absence (−) or presence (+) of reducing agent DTT). The black arrow indicates the bands of the disulfide stabilized sE dimer.

(SDS–PAGE) under reducing and non-reducing conditions (Fig. 3e). We also prepared the mutants equivalent to DENV2 A259C and L107C/A313C for the other DENV serotypes where we were obtaining similar yields, except for DENV1, which resulted only in monomer or aggregates with little dimer formation (Supplementary Fig. 3), indicating that in the context of the DENV1 E protein sequence, the polypeptide chain cannot adapt to the correct geometry to make the disulfide

bond, in contrast to the E protein from the other DENV serotypes.

**Structure of dimer DENV2 sE mutants/EDE2 A11 Fab complexes.** To confirm that inter-subunit disulfide bond formation did not interfere with the overall conformation of the sE dimer, we carried out structural studies of the DENV2 sE mutants by X-ray crystallography. Because the mutants by themselves did not yield crystals of good enough quality, we tested crystallization in complex with EDE mAb fragments. We obtained crystals of both sE A259C and sE L107/A313C in complex with Fab EDE2 A11 belonging to the orthorhombic space group $P2_12_12$. The crystals diffracted anisotropically to 3.9 Å along the *h* and *l* axis and to only 7.2 Å (A259C) or 6.2 Å (L107/A313C) along the long *k* axis. Given the limited resolution and anisotropy of the datasets we carefully selected the resolution cutoffs leading to the statistics shown in Table 1.

Both mutants crystallize with near identical unit cell parameters as the wild-type DENV2 sE-Fab EDE2 A11 complex (PDB accession code 4UTB,[22]), but in a different orthorhombic space group ($P2_12_12$ for the mutants and of $P2_12_12_1$ for the wild type), resulting in a related but non-identical crystal packing. This lack of isomorphism precluded the calculation of an isomorphous difference map ($Fo_{Mutant}-Fo_{WildType}$) with the 4UTB structure. We therefore determined the crystal structures of the mutants by molecular replacement using the 4UTB model as search. The resulting difference electron density maps, after one round of refinement, showed strong positive peaks mapping to the sites where the disulfide bonds were introduced (Supplementary Fig. 4a).

Refinement of the atomic models of the mutants was done at 3.9 Å resolution avoiding over-fitting the data as much as possible by keeping very tight geometric constraints (Table 1). The results are consistent with the introduction of the disulfide bonds in both sE dimer mutants not inducing a gross rearrangement of the global conformation of the sE dimers, as illustrated in the superpositions displayed in Supplementary Fig. 5, with a root mean square deviation of 1.11 Å for the A259C mutant and 0.14 Å for the L107C-A313C mutant over 775 superposed Cα atoms. The final refined structures of sE mutants thus have the same overall organization as wild-type sE (Fig. 4), except for the extra disulfide bond located at the two-fold axis for the sE A259C structure (green arrow in Fig. 4b), and the two extra disulfide bonds connecting the fusion loop with domain III (green arrow in Fig. 4c) for the sE L107C/A313C structure. The resulting electron density maps obtained with the final refined phases (2Fo-Fc map) around the inter-chain disulfide bonds for sE A259C and sE L107C/A313C mutants are displayed in Supplementary Fig. 4b.

**Two disulfide bridges suppress exposure of the FLE.** To verify that EDE epitopes are effectively exposed in our mutant sE dimers, we tested binding to both anti-EDE1 and anti-EDE2 mAbs in a direct ELISA assay (as described in Fig. 1a) with plate

## Table 1 | Data collection and refinement statistics.

| | DENV2 sE A259C–EDE2 A11 Fab fragment (PDB 5N0A) | DENV2 sE L107C/A313C–EDE2 A11 Fab fragment (PDB 5N09) |
|---|---|---|
| *Data collection* | | |
| Space group | P 2₁ 2₁ 2 | P 2₁ 2₁ 2 |
| Cell dimensions | | |
| a, b, c (Å) | 182.3, 208.7, 58.8 | 180.6, 205, 58.9 |
| α, β, γ (°) | 90, 90, 90 | 90, 90, 90 |
| Resolution (Å) | 50 − **3.9** (4.1 − 3.9)* | 50 − **3.9** (4.3 − 3.9)* |
| | | |
| *Anisotropy directions* | | |
| Resolution where CC$_{1/2}$ > 0.30 | | |
| overall (Å) | **3.9** | **3.9** |
| along *h* axis (Å) | **3.9** | **3.9** |
| along *k* axis (Å) | 7.2 | 6.2 |
| along *l* axis (Å) | **3.9** | **3.9** |
| Rmerge | 0.31 (0.84) | 0.69 (1.97) |
| Rmeas | 0.41 (1.08) | 0.84 (2.42) |
| Rpim | 0.26 (0.68) | 0.48 (1.4) |
| $<I/\sigma(I)>$ | 3.1 (1.4) | 2.4 (0.7) |
| Mn(I) half-set correlation | 0.95 (0.68) | 0.89 (0.51) |
| Completeness (%) | 96.9 (97.5) | 96.9 (98.0) |
| Redundancy | 3.8 (3.8) | 5.1 (4.9) |
| | | |
| *Refinement* | | |
| Resolution | 20.0 − 3.9 (4.1 − 3.9)† | 40.0 − 3.9 (4.1 − 3.9) |
| No. of reflections (Work/Test) | 19,112/947 | 19,323/945 |
| $R_{work}$ / $R_{free}$ | 0.29/0.30 (0.27/0.31) | 0.31/0.35 (0.37/0.38) |
| No. of atoms (protein) | | |
| Protein | 9542 | 12683 |
| Glycans | 143 | 157 |
| Water | 0 | 2 |
| R.m.s deviations | | |
| Bond lengths (Å) | 0.007 | 0.005 |
| Bond angles (deg) | 1.13 | 1.20 |

CC$_{1/2}$, correlation coefficient; PDB, Protein Data Bank; Rmeas, multiplicity-corrected R; Rpim, expected precision.
The high-resolution limit used for refinement after anisotropy statistics is shown in bold.
*Highest resolution shell is shown in parenthesis.
†Low-resolution for refinement of DENV2 sE A259C–EDE2 A11 Fab complex was truncated to 20 Å.

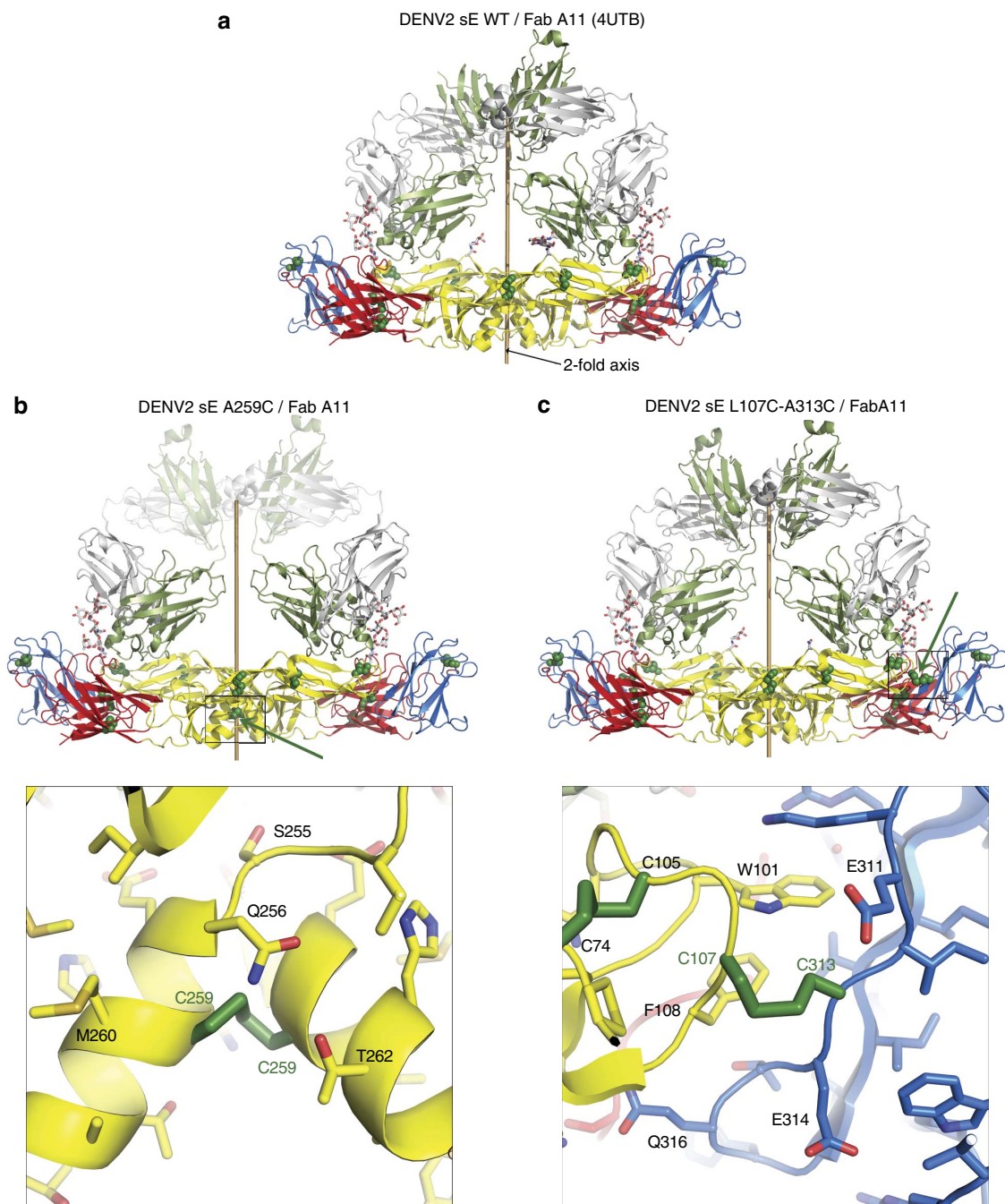

**Figure 4 | Structures of DENV2 sE FGA02 WT and mutants in complex with anti-EDE2 Fab A11.** (**a**) The previously determined structure of DENV2 sE WT in complex with EDE2 A11 Fab (PDB code 4UTB). The molecular two-fold axis is shown as a light-brown central rod, and the cysteines are displayed as green spheres. The heavy and light chains of Fab A11 are coloured in green and light grey, respectively. sE proteins are colour-coded by domains: domain I–red, domain II–yellow and domain III–blue. (**b,c**) Structures determined here of DENV2 sE A259C mutant in complex with anti-EDE2 A11 Fab (**b**) and DENV2 sE L107/A313C mutant in complex with anti-EDE2 A11 Fab (**c**). The constant domains of the two Fab A11 in DENV2 sE A259C complex were disordered in the final structure and thereby are shown in transparent ribbons. Lower **b,c**: zoom views of the engineered disulfides are shown respectively for DENV2 sE A259C in complex with anti-EDE2 A11 Fab and DENV2 sE L107/A313C in complex with anti-EDE2 A11 Fab.

bound recombinant wild type sE protein in comparison to covalently linked sE dimers performed at low E concentration (10 µg ml$^{-1}$) (Fig. 5a,b left panels). Both A259C (Fig. 5a) and L107C/A313C (Fig. 5b) mutants were efficiently recognized by anti-EDE1 and anti-EDE2 antibodies, in contrast to the wild type protein and in line with the crystal structures, showing that the mutant recombinant sE proteins form stable dimers correctly exposing the EDE.

As the FLE is an immunodominant epitope, it is important to design immunogens that prevent the generation of this suboptimal ADE inducing response. To assess whether the stable sE-dimers described above were still capable of exposing the FLE, we tested their reactivity to a panel of FLE reactive human mAbs we have previously described[12].

Binding to the panel of anti-FLE mAbs was tested to sE-dimers and compared to binding to wild type sE (Fig. 5a,b right panels).

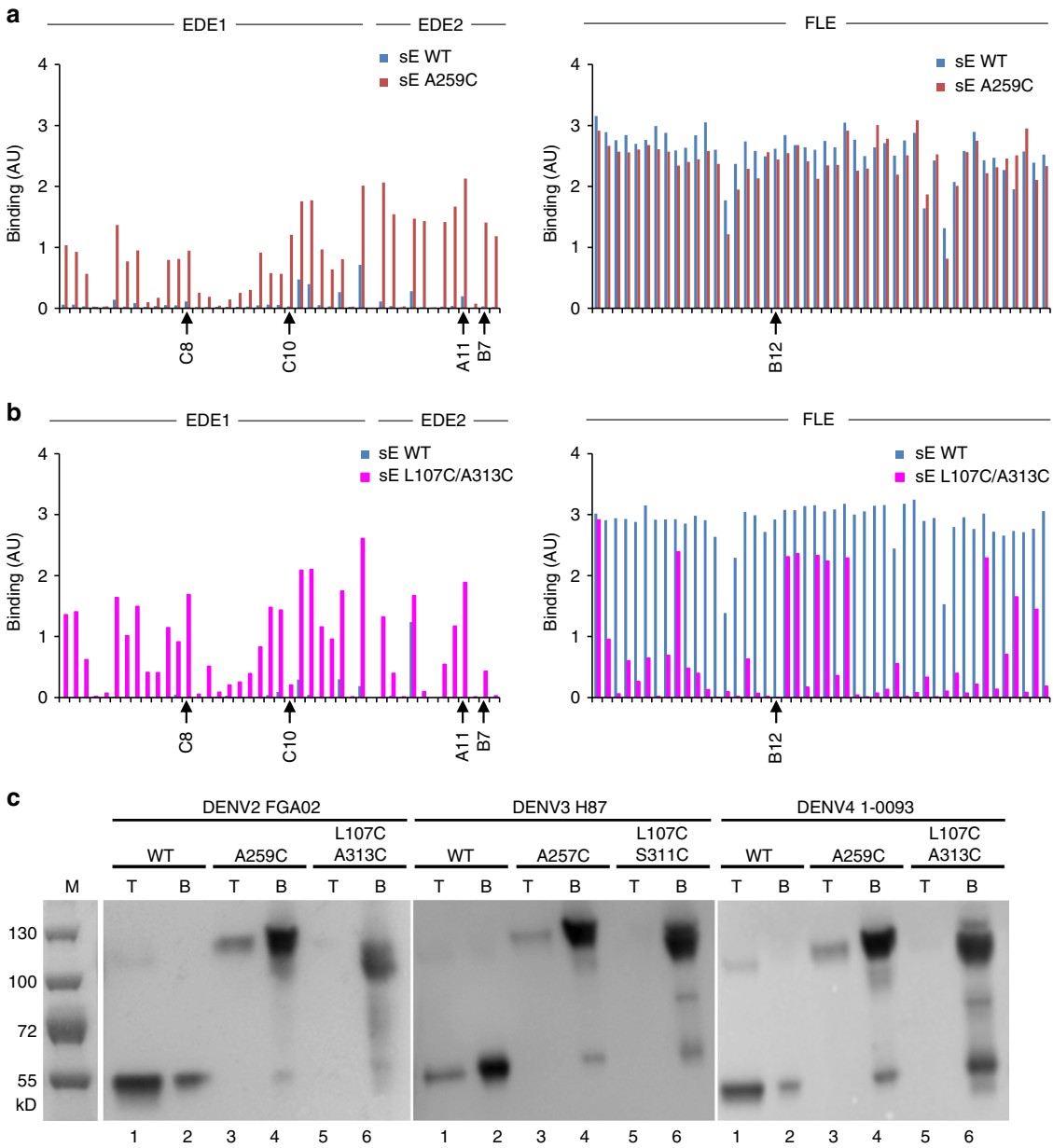

**Figure 5 | Covalently linked sE dimers recapitulate the EDE and do not interact with liposomes.** (**a,b**) ELISA plates were coated with DENV2 either wild type monomeric sE (sE WT) or the two covalently linked sE dimers (**a**) A259C or (**b**) L107C/A313C and following incubation with panel of anti-EDE1 and anti-EDE2 mAbs (left panel) or anti-FLE at $1\,\mu g\,ml^{-1}$ (right panel), binding was determined using ALP-conjugated anti-human IgG. (**c**) Results of co-flotation with liposomes in an Optiprep gradient at low pH (see Methods for lipid composition). Wild type, single and double mutants from DENV2 FGA02, DENV3 H86 and DENV4 1-0093 were incubated at pH 5.8 with liposomes and run in an Optiprep gradient. Insertion of the WT sE proteins in the liposome membrane results in its floatation to the low density top (T) fractions of the gradient (lanes 1). A fraction of the single mutants appears to still be able to float with the liposomes (lanes 3), whereas in the double mutants, there is no sE protein recovered from the top fraction (lanes 5), in line with the fact that the FLE is not exposed.

Of note, we found that the single disulfide bonded A259C sE dimer was fully competent to bind the panel of anti-FLE mAbs (Fig. 5a, right panel), whereas binding was largely lost when using the double disulfide bonded L107C/A313C sE dimer (Fig. 5b, right panel). We surmise that A259C mutant, which is linked via a single bond at the centre of the sE dimer (Fig. 4b), shows a considerable dynamic behaviour, being able to breathe by swivelling around the central disulfide bond and allowing

exposure of the FLE. On the other hand, sE dimer L107C/A313C, containing disulfide bonds at each end of the dimer (Fig. 4c), is locked and unable to expose the FLE.

To further confirm this hypothesis, we checked the ability of these two mutants to insert into membranes using a liposome floatation assay. DENV2 WT, A259C and L107C/A313C recombinant proteins were incubated with liposomes under acidic conditions and analysed by western blot after ultra-

centrifugation on a density gradient (Fig. 5c). Under acidic conditions WT sE inserts into the liposomes and the majority of the protein is found in the top fraction of the gradient (Fig. 5c lanes 1). Mutant A259C was still able to insert into liposomes, as it could be detected in the top fraction of the gradient (Fig. 5c lanes 3)—albeit proportionally less than wild type—whereas no protein co-floated with the liposomes in the L107C/A313C mutant (Fig. 5c lanes 5). These results suggest that in the A259C mutant the fusion loop is partially exposed and is able to insert into membranes as dimer, making weaker interactions than wild-type, which inserts as trimer[27]. This result is in line with the ability of FLE mAbs to bind A259C mutant in the ELISA assay (see above Fig. 5a), and suggests that the two monomers can rotate about the engineered disulfide bond to align parallel to each other for membrane insertion. In the L107C/A313C mutant the fusion loop is not exposed to promote insertion into membranes under acidic conditions, in concordance with reduced FLE exposure shown in Fig. 5b. Similar results were obtained for DENV3 and DENV4 (Fig. 5c).

## Discussion

We have shown here that it is possible to engineer covalently locked sE dimers that expose the EDE, and are not recognized by most antibodies targeting the FLE. Such stabilized sE dimers can be made in high yields for three out of the four DENV serotypes, with the exception of DENV1. We will attempt to express DENV1 dimers in alternative expression systems, however, as the EDE is a cross-reactive epitope, the EDE mAbs efficiently neutralize DENV1, as well as the other serotypes. In the future, priming and boosting by alternating the three DENV serotypes for which the dimers can be obtained should be sufficient to preferentially stimulate B cells producing EDE antibodies, which by definition would also neutralize DENV1. The possibility remains to resurface sE from DENV3 (which is most closely related to DENV1) in the event that a DENV1-like sE molecule with a stable EDE is required, a similar approach has recently been used to graft the DENV3 specific epitope for mAb 5J7 into DENV1 (ref. 28).

It is now becoming clear that the most potent neutralizing antibodies against DENV target conformationally sensitive epitopes readily exposed at the E dimer surface. Some of these epitopes are contained within a single E subunit, such as the potently DENV4-serotype specifc mAb-5H2 or the DENV1 specific mAb-1F4, and some are shared by two or more E subunits such as the DENV1 specific mAb-HM14c10 or the DENV2 specific 2D22-mAb, which binds the E-dimer at a position close to the EDE, or the DENV3-specific mAb-5J7, which binds across three adjacent E polypeptides on the virion[13,15,29–31].

prM-specific antibodies do not bind to fully mature virions—since they do not contain prM—whereas many partially mature particles do not contain a high enough density of prM to allow neutralization but yet may be sufficient to promote ADE[20]. We have speculated that the inefficient cleavage of prM may be an immune evasion/enhancement strategy (indeed, the furin cleavage sequence is suboptimal in DENV prM), leading to the generation of poorly neutralizing antibodies directed to prM. The high frequency, low potency and high ADE potential of antibodies directed to prM has implications for vaccine design; all attenuated vaccines constructs encode prM, although the precise prM content of the virus particles in these vaccines has not been reported. The ideal vaccine would focus responses to E, while the prM component of the response be minimized if the potential for ADE in vaccines is to be reduced. The stabilized sE-dimers described here are devoid of prM and will therefore not induce this response.

Compared to other flaviviruses such as ZIKV, DENV seems to display a higher dynamic behaviour, which combined with incomplete furin maturation allows for exposure of the FLE in particles circulating at neutral pH (refs 32–37). It is highly likely that the dynamic behaviour of DENV particles may underpin why the FLE is such a dominant epitope in DENV and consequently why it is difficult to produce effective DENV vaccines. Our demonstration that engineering inter-subunit disulfide bonds that do not alter the structure of the E dimer (Fig. 4) and are not recognized by anti-FLE mAbs (Fig. 5) is an important step towards avoiding elicitation of antibodies targeting the FLE. We propose that future vaccines minimizing the anti-FLE response should be pursued as immunogens. The double disulfide bonded DENV2 L107C/A313C dimer described here, where the dimer is effectively locked and does not expose the FLE, is a strong candidate to constitute the basis for such a vaccine. Testing the ability of DENV stabilized dimers to elicit an anti-EDE response in mice may be complicated as murine antibody responses differ considerably from human responses in particular murine responses are much more directed to domain III of E and complex conformational epitopes appear rare. Furthermore, we anticipate focussing the response to the EDE may require heterologous prime boosting strategies.

Dengue vaccines are now at an important juncture; a large scale Phase III trial has underperformed expectations and given a concerning safety signal of enhanced infection[5,38]. Here we have demonstrated the feasibility of locking the E-dimer to avoid generation of poorly neutralizing antibodies such as those targeting immunodominant FLE. Together with the elimination of prM from an appropriate candidate subunit vaccine, this approach has the potential to generate broadly protecting subunit DENV/ZIKV vaccines.

## Methods

**Cells and Reagents.** The C6/36 cell line (a gift from Dr Malasit, Mahidol University Thailand) derived from mosquito *Aedes albopictus* was cultured in Leibovitz L-15 at 28 °C. Vero and 293 T cells (gifts from Dr Malasit, Mahidol University Thailand) were grown at 37 °C in MEM and DMEM, respectively. All media were supplemented with 10% heat-inactivated fetal bovine serum (FBS), 100 units per ml penicillin, 100 µg ml$^{-1}$ streptomycin and 2 mM L-Glutamine. Monocyte-derived dendritic cells (MDDCs) were prepared as previously described[39]. Brifely, CD14$^+$ cells isolated from PBMC were differented to dendritic cells by culturing in RPMI1640 containing 10%FBS, 25 ng ml$^{-1}$ of rIL-4 and 20 ng ml$^{-1}$ of rGM-CSF for 5–6 days. Cells were then checked for upregulation of DC markers including DC-SIGN, CD80 and CD86. All cell lines were free from mycoplasma contamination.

2C8, a murine mAb directed to envelope domain III of DENV2 (EDIII), was a gift from Dr C. Puttikhunt and Dr W. Kasinrerk (Biotec and Chiang Mai University Thailand). Alkaline phosphatase (ALP)-conjugated anti-human IgG (A9544), ALP-conjugated Streptavidin (S2890), p-nitrophenylphosphate (PNPP, N2770-50) and Bovine serum albumin (BSA, A7030) and polyethylenimine (408727) were purchased from Sigma. Strep Tactin-ALP conjugate (2-1503-001) was from IBA GmbH. Goat anti-mouse Igs-HRP (P0447) and Rabbit anti-human IgG-HRP (P0214) were from Dako. Luminata Classico Western HRP Substrate (WBLUC0100) was from Merck Millipore. MEM (31095) and Leibovitz L-15 (11415) were from Gibco, RPMI (R8758) and DMEM (D6046) were from Sigma and UltraDOMA-PF (12-727F) was from Lonza. rHu GM-CSF (62-10-209) was from First Link (UK) Ltd. and rHu IL4 (34-8049) was from eBioscience.

**Virus stock.** Dengue virus serotype 2 (16681) (a gift from Dr Malasit, Mahidol University Thailand) were grown in C6/36 cells and MDDCs. Cell-free supernatants were collected and stored at −80 °C. The titres of virus were determined by a focus forming assay on Vero cells and expressed as focus-forming units (FFU) per ml (ref. 40).

**Expression of human monoclonal anti-DENV E antibodies.** A pair of plasmids containing the heavy and light chains of human IgG[12] were co-transfected into 293 T cells using the polyethylenimine method and cultured in protein-free media (UltraDOMA-PF). Culture supernatant containing antibodies was collected after 5 days. The abbreviation for anti-EDE1 mAbs 752-2C8 and 753(3)C10 were C8 and C10, respectively[12]. The abbreviation for anti-EDE2 mAbs 747(4)A11 and

747(4)B7 were A11 and B7, respectively[12]. The abbreviation for anti-FLE mAb 749B12was B12 (ref. 12).

**sE direct ELISA.** Purified DENV2 sE strain FGA02 (GenBank accession number KM087965.1) (WT, A259C or L107C/A313C) and BSA were used as coating antigen and negative control antigen, respectively. NUNC immobilizer plates (436006) were coated with 50 µl of 10 µg ml$^{-1}$ protein and blocked with 3% BSA. Plates were then incubated with 50 ul of 1 µg ml$^{-1}$ of anti-FLE and anti-EDE mAbs supernatants followed by ALP-conjugated anti-human IgG. The activity was observed with PNPP and measured at 405 nm.

**Determination of the ability of anti-EDE mAb to stabilize the E-dimer.** A MAXISORP immunoplate (442404; NUNC) was coated with 50 ul of 5 µg ml$^{-1}$ of human anti-FLE or anti-EDE mAbs. The plate was then blocked with 3% BSA for an hour followed by incubation with serial dilutions of Strep-tagged recombinant envelope protein DENV2. The reaction was visualized by ALP-labelled Strep-Tactin and PNPP substrate. The reaction was stopped by adding NaOH and the absorbance was measured at 405 nm.

**Biochemical analyses.** SEC-MALS was performed by loading ∼150 ug of DENV2 FGA02 A259C protein into Superdex 200 10/300 GL column (GE life sciences) and samples were run in Tris 50 mM, NaCl 500 mM (pH 8.0) at a flow rate of 0.4 ml min$^{-1}$. These samples passed through a Wyatt DAWN Heleos II EOS 18-angle laser photometer coupled to a Wyatt Optilab TrEX differential refractive index detector. Data were later analysed using Astra 6 software (Wyatt Technology Corp).

Analysis of the complex of DENV2 FGA02 WT along with Fab C8 and Fab A11 were performed in similar manner by loading 150 µg of DENV2 FGA02, 300 µg of Fab C8, 300 µg of Fab A11 and for complex formation a mixture of 150 µg of DENV2 FGA02 with 300 µg of FabC8 and 150 µg of DENV2 FGA02 with 300 µg Fab A11. Proteins were injected using 100 µl loop.

**Ab competition ELISA.** For mAb competition assays, DENV2 produced from C6/36 cell lines or MDDCs were captured onto MAXISORP immunoplates coated with 2C8 (a gift from Dr C. Puttikhunt and Dr W. Kasinrerk Biotec and Chiang Mai University Thailand); a mouse anti-EDIII mAb specific for DENV2, plates were then blocked with 3% BSA. An equal volume of fixed concentration of biotin labelled anti-FLE or anti-EDE mAbs at 1 µg ml$^{-1}$ was mixed with a serial dilution of unlabelled anti-FLE, anti-EDE or irrelevant anti-Flu mAbs. The mixtures were then added to DENV captured to ELISA plates and incubated for 1 h, following washing plates were then incubated with ALP-conjugated Streptavidin. The reaction was developed by the addition of PNPP substrate and stopped with NaOH. The absorbance was measured at 405 nm. For competitive ELISA, the signals are inversely proportional to the ability of unlabelled mAbs to compete with the biotin-labelled mAb for binding.

For the Ab displacement ELISA, biotin-labelled anti-FLE or anti-EDE mAbs was first added onto DENV captured to ELISA plates and incubated for 1 h. After washing, unlabelled anti-FLE, anti-EDE or irrelevant anti-Flu mAbs (a gift from Dr Xu Imperial College London UK) was sequentially added and incubated for further 1, 5, 15, 30 and 60 min. The reaction was then developed by adding ALP-conjugated Streptavidin and PNPP substrate. The ability of second unlabelled mAb to displace the binding of first biotin-labelled will yield a lower signal.

**Measurement of prM cleavage on DENV.** The efficiency of PrM cleavage was evaluated by running viral supernatants from C6/36 cells and MDDCs on 12% SDS–PAGE and western blotting with mouse anti-E mAb (4G2) (a gift from AFRIMS Thailand) and human anti-prM (3–147)[20] followed by a cocktail of goat anti-mouse Igs and rabbit anti-human IgG. Finally, the membrane was developed with chemiluminescence substrate. In addition, the levels of prM cleavage were also analysed by detection of E and prM by ELISA[20]. Briefly viral supernatants from C6/36 and MDDC cells were captured onto plates coated with murine anti-E mAb (4G2). Then, E and prM were detected using a humanized version of 3H5 mAb (hu3H5) (a gift from AFRIMS Thailand) and human anti-prM (3–147), respectively.

**Production and purification of wild type and disulfide stabilized sE proteins.** Recombinant sE from different serotypes were cloned, as previously described[22]. Single and double cysteine mutations were introduced by standard Gibson assembly cloning with primers containing mutant sequence and the mutation in constructs was verified by DNA sequencing. All the constructs were transfected into Drosophila S2 cells (ThermoFisher Scientific, R690-07) and expressed, as described previously[22]. All of the mutants, as well as the WT protein produced different amounts of soluble aggregates that was removed by SEC. DENV2 WT and DENV2 A259C proteins were purified using affinity streptactin followed by SEC in 50 mM Tris (pH 8) and NaCl 500 mM. However, double disulfide bonded DENV2 L107C/A313C was observed to produced large amounts of soluble aggregate along with the dimer peaks in SEC after affinity chromatography and additional

purification steps were introduced. DENV2 L107C/A313C mutant protein peak collected after affinity streptactin was adjusted to pH 8.5 with 100 mM Tris and to 2.5 M NaCl and bound to 1 ml hydrophobic interaction phenyl column. Protein was eluted using slow NaCl gradient from 2 M NaCl to 0 mM NaCl (30 column volumes). Mutant dimer protein eluted at 65.2 mS ml$^{-1}$ conductance value in phenyl column purification conditions. This protein was further purified using SEC (in 50 mM Tris pH 8 500 mM NaCl).

DENV3 WT, DENV3 A257C and the DENV4 WT, DENV4 A259C mutant proteins were purified using the same protocol, as for DENV2 WT and A259C stated above. DENV3 L107C/S311C and DENV4 L107C/A313C were purified using streptactin affinity chromatography followed by SEC purification (in 50 mM Tris pH 8, 200 mM NaCl) SDX200 column, where the aggregates and monomer peaks were separated from the dimer peak. The dimer peak from both was subjected to another SEC (in 20 mM NaH2PO4, 250 mM NaCl and 20 mM disodium succinate, pH 7) for polishing. These purified proteins were buffer exchanged into Tris 50 and 500 mM NaCl and stored or used for assays. Yields of different proteins per liter of S2 cell culture are provided in Supplementary Fig. 3a.

**Crystallization and three-dimensional structure determinations.** The optical density at 280 nm of each protein solution was measured before crystallization and was 0.6 and 0.4 for DENV2 sE-A259C/EDE2 A11 Fab and DENV2 sE-L107C/A313C-EDE2 A11 Fab complexes, respectively. The protein concentration estimated using theoretical extinction coefficients for the individual components, DENV2 sE-strep and EDE2 A11 Fab, was 1.03 and 1.68, respectively, without taking into account carbohydrate moieties. The protein buffer used for all the crystallization experiments was 150 mM NaCl and 15 mM Tris pH 8. Crystallization trials were performed in sitting drops of 400 nl at 18 °C. Drops were formed by mixing equal volumes of the protein and reservoir solution in the format of 96 Greiner plates, using a Mosquito robot, and monitored by a Rock-Imager. Crystals were optimized with a robotized Matrix Maker and Mosquito setups on 400 nl sitting drops, or manually in 24-well plates using 2–3 µl hanging drops at 18 °C. The crystals of DENV2 sE A259C in complex with EDE2 A11 Fab were grown in 100 mM Hepes pH 7.5, 19% PEG 6 K and 1.5% (v/v) MPD. The crystals of the complex DENV2 sE L107C/A313C with EDE2 A11 Fab were obtained in 100 mM Tris pH 8.5 and 20.7% PEG 4 K. The crystals of A259C and L107C/A313C mutants were transferred in drops containing 67% of mother liquor with respectively 16% of glycerol and 16% of ethylene glycol as cryoprotectants, before being cryo-cooled in liquid nitrogen. X-ray diffraction data were collected at beam lines PROXIMA-2 at the SOLEIL synchrotron (St Aubin, France), and ID29 at the European Synchrotron Radiation Facility (Grenoble, France). Only one crystal was used for each of the data sets. Diffraction data were processed using the XDS package and scaled with SCALA or AIMLESS[41] in conjunction with other programs of the CCP4 suite[42]. The high resolution limits for each structure were determined using CC$_{1/2}$-based cutoffs of 0.30 (ref. 43). The structures were determined by molecular replacement with PHASER[44] using the search model DENV2 sE WT in complex with EDE2 A11 Fab (PDB 4UTB).

Subsequently, careful model building with COOT[45] was done in 2Fo–Fc electron density map (Supplementary Fig. 6), alternating with cycles of crystallographic refinement with the programs Phenix.Refine[46] and/or BUSTER/TNT[47], led to the final model. Refinement was constrained to respect non-crystallographic symmetry with TLS refinement[48] and target restraints using higher-resolution structures of DENV2 sE (from PDB 4UTA), scFv A11 (PDB 4UT7) and constant domains of Fab A11 (from PDB 4LLD). Electron density sharpening maps were computed with COOT, and helped for the manual model building[49]. Refined crystallographic models were analysed with MolProbity[50], which indicated that 99.67% and 99.82% of residues were in allowed conformations, respectively for A259C and L107C/A313C structures. The figures were prepared using the PyMOL molecular Graphics System (Schrodinger)(pymol.sourceforge.net).

**sE-liposomes co-flotation assay.** Liposomes were prepared by freeze-thaw and extrusion through 100 nm pore size polycarbonate filters (Whatman 800309) using a 1:1:1:3 molar ratio of DOPC (1,2-dioleoyl-sn-glycero-3-phosphocholine) (850375C), DOPE (1,2-dioleoyl-sn-glycero-3-phos- phoethanolamine) (850725C), sphingomyelin (from bovine brain) (860062C), cholesterol (from ovine wool) (700000P) purchased from Avanti Polar Lipids. Purified sE (WT, A259C or L107C/A313C from DENV2 1-0093 or DENV4 and WT, A257C, L107C/S311C from DENV3 H87) was mixed with liposomes and incubated for 10 min at RT before overnight incubation at 30 °C under acidic conditions. The mixture was separated by ultracentrifugation on an Optiprep (Proteogenix 1114542) continuous 0–30% gradient. Aliquots from top and bottom fractions were analysed by western blot using an anti-strep tag antibody.

**Data availability.** Coordinates and structure factors amplitudes have been deposited in the Protein Data Bank under accession numbers 5N0A and 5N09 for DENV2 sE A259C and DENV2 sE L107C/A313C mutants in complex with EDE2 A11 Fab complexes, respectively. All other relevant data are available from the authors on request.

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

## Acknowledgements

We thank Watchara Kasinrerk and Chunya Putthikhunt for anti-dengue anti-DomIII mAb 2C8; J. Freire and G. Bowler for help and discussion; the staff at the crystallogenesis and chemogenomic & biological screening facilities at Institut Pasteur; the staff at beamlines PX1 and PX2 at SOLEIL synchrotron (Saclay, France), the staff at beamlines ID23-1, ID23-2, ID29 and IB30B at the European Synchrotron Radiation Facility (Grenoble, France); Fabrice Agou at Institut Pasteur for the MALS system. We acknowledge support from the European Commission FP7 Programme for the DEN-FREE project under Grant Agreement number 282 378FP7 (F.A.R., J.M. and G.R.S.); the 'Integrative Biology of Emerging Infectious Diseases' Labex (Laboratoire d'Excellence) grant number ANR-10-LABX-62-IBEID (French Government's 'Investissements d'Avenir' program ) (F.A.R.) the National Institute for Health Research Biomedical Research Centre, Funding Scheme, UK (G.R.S.); and the NEUTRAVIR grant from Région Ile-de-France (DIM-Maladies Infectieuses) (F.A.R.). G.R.S. is a Wellcome Trust Senior Investigator.

## Author contributions

F.A.R., G.R.S. and J.M. designed the experiments, P.G.-C., F.A.R. and A.R. designed the cysteine mutants; A.R. and A.S. produced and purified the recombinant DENV2-3-4 sE proteins and the antibody fragment A11; W.D., P.S. and W.W. produced antibodies and performed binding experiments; A.H., A.R. and M.-C.V. grew and optimized the crystals. M.-C.V., A.R., P.G.-C., S.D. and A.S. collected synchrotron data; M.-C.V., P.G.-C. and S.D. processed the data, built, refined and analysed the atomic models. A.R. and P.G.-C. made the MALS experiments; G.B.S. did the floatation experiments with liposomes. G.R.S. and F.A.R. wrote the paper with the help of J.M., G.B.-S., A.R., M.-C.V., W.D. and P.G.-C.

## Additional information

**Competing interests:** The EDE antibodies, EDE epitope and envelope protein dimers that induce EDE antibodies are the subject of a patent application by Imperial College and Institute Pasteur on which G.S., J.M., F.A.R., A.R., G.B.-S., P.G.-C., M.-C.V. and S.D. are named as inventors. The remaining authors declare no competing financial interests.

