## [Peer Review File · Nature Communications]

Reviewers' comments:

Reviewer #1 (Remarks to the Author):

Rouvinski et al, attempts to produce a dengue E protein vaccine candidate that contains epitope that spans across the E proteins within a dimer and also at the same time, had their fusion loop hidden. This is because antibodies that binds across E protein (aka EDE antibodies) had been shown to be highly protective against the diseases, whereas those binding to fusion loop enhances the pathology of the disease. The authors generated different stable E protein dimers by making mutations on the E proteins leading to formation of disulphide bridge between the protomers. They characterized these mutants and showed that their structures are similar to the wild type E protein dimer, allowing EDE antibodies to bind. One of these mutants, the L107C/A313C, seemed to have successfully hid their fusion loops, therefore suggesting that this would be a strong vaccine candidate. The work is significant and important as it shows the possibility of making a construct that may elicit antibodies that bind quaternary structure dependent epitope, although the efficacy of this E protein dimer vaccine candidate is yet to be proven. The paper is overall very well written and easy to read.

Comments:

(1) Page 5 line 162: "C6/36 produced viruses have a much higher content of uncleaved prM compared to DENV produced in DC."

In other labs, some virus preparations from C6/36 can have low prM content, it could be a matter of prep to prep differences. Since the assumption of the high content of prM in your C6/36 is important for the interpretation of the ELISA competitive results, the authors should show a gel of the prM content of the virus that were used for these assays in the supplementary figures.

(2) For figure 3a, the structures scored B, C D and D down the panel, respectively. Why is there is no structure that scored A?

(3) Page 7, line 203, L017C should be L107C

(4) Page 8, line 244, "The final refined structures of sE mutants have the same overall organization as wild-type sE...". Authors should provide a figure that superimposed the mutants with the WT E protein to show the similarities and also provide RMSD values.

(5) Page 9, last paragraph of the result section, authors suggests that A259C mutant has their fusion loop exposed and therefore able to insert into the liposome. However, the gel shows that most of the A259C are still at the bottom fraction and only some are at the top, unlike the WT DENV2 and DENV4 E proteins which showed more on the top fractions. Therefore, the paragraph should be toned down to:

"Some mutant A259C was still able to insert into liposomes and could be detected in the top fraction of the gradient (Fig. 6 lanes 3) whereas no protein co-floated with the liposomes in the L107C/A313C mutant (Fig. 6 lanes 5). These results suggest that in the A259C mutant the fusion loop is partially exposed and some of the proteins are able to interact with membranes in spite of the covalent linkage that prevents the sE protein from transitioning

to the trimeric state and in line with the ability of FLE mAbs to bind A259C mutant in the ELISA assay (see above Fig. 5a)."

(6) Page 11 line 349, "all attenuated vaccines contain prM" – has this been determined? I would recommend deleting this unless there are publications or strong evidence on this.

(7) Supplementary Figure 2b. DENV2 coomassie stained gel is not shown.

Reviewer #2 (Remarks to the Author):

In this manuscript from Rouvinski and colleagues evaluate the possibility of using dengue virus engineered E protein dimers as immunogens in next generation vaccines. The authors compare the activities of two classes of neutralizing dengue antibodies: those that recognize the immunodominant fusion loop epitope (anti-FLE) and those that recognize the E dimer epitope anti-EDE). They elegantly show that anti-FLE bind to both monomeric and dimeric forms of E whereas anti-EDE bind to dimeric E and are potent and broadly neutralizing. They introduced cysteine substitutions that act to crosslink the E protein dimers and show that these are unaffected in structure and can be readily recognized by the two types of anti-EDE antibodies. Importantly, using a double crosslinked E dimer, but not a single, they show that the immunodominant FLE is no longer accessible. The authors argue that such a crosslinked dimer could be effectively used to direct the immune system toward a more potent, neutralizing antibody response, with a lower probability of eliciting ADE. The results presented are important and the experiments well executed. This manuscript represents an intellectual advance in the field and offers alternative considerations for next generation dengue virus vaccines.

Comments:

Figure 4: It is difficult to see the sites for cysteine substitution. Perhaps a blow up of this region in panels b and c would improve visualization of these sites.

Line 277: Although this floatation assay is provided to support their hypothesis that the single crosslink remains dynamic whereas the double substitution locks down the fusion loop exposure, can the authors provide any structural or biophysical evidence for this hypothesis? What about B factors from the two structures?

Discussion: The Discussion on page 10 reads like a review and should be truncated.'

Supplementary Table 1: The PDB coordinates need to be added.

Reviewer #3 (Remarks to the Author):

In this study Rouvinski et al describe the structure and antibody binding properties of variants of the dengue virus envelope glycoprotein, E, which were engineered to be

constitutively dimeric through the introduction of cysteine residues that form intermolecular disulfide bonds across the E dimer interface. The authors show that most antibodies against the immunodominant fusion loop epitope (FLE)- 36 out of 46 tested- fail to bind a soluble E fragment with two engineered intermolecular disulfides (sE L107C/A313C) in an ELISA. In contrast, most antibodies against the E dimer epitope (EDE)- 28 out of 42 tested- were able to bind the sE L107C/A313C variant. This stands in sharp contrast to the wild type sE protein, which binds all 46 anti-FLE antibodies and only one or two of the EDE antibodies. This is significant because anti-EDE antibodies were shown by a similar group of authors to be broadly neutralizing against all four dengue serotypes, whereas anti-FLE antibodies can be potentially harmful as they are often poorly neutralizing, potentially induce antibody-dependent enhancement of infection, and are the most type of antibody produced most abundantly during dengue virus infection. Hence the authors claim that stabilized E dimers such as sE L107C/A313C could serve as a broadly protecting subunit vaccine against dengue virus, and also against Zika virus, in which many of the EDEs are conserved.

These key findings and central claim are underpinned by various supporting data including: (1) biochemical studies showing that anti-EDE antibodies stabilize wt sE and that the sE L107C/A313C variant is a dimer in solution, (2) crystal structures of sE L107C/A313C and another stabilized sE variant in complex with an EDE antibody showing that the EDE epitopes are intact in the sE variants, (3) a competition ELISA showing that dengue virus particles produced mosquito cell (which have more exposed FLE epitopes than viruses produced in human cells) preferentially bind to anti-FLE antibodies rather than anti-EDE antibodies when both are present, and (4) sE L107C/A313C variants from dengue serotypes 2, 3 and 4 did not interact with liposomes, whereas the wild type and single-disulfide variants did, confirming that the fusion loop (and FLE) is not exposed in the sE L107C/A313C.

This manuscript is well written, timely and overall quite convincing. The authors showed previously that anti-EDE antibodies broadly and potently neutralize dengue viruses from all four serotypes. Eliciting this type of antibody during infection is therefore desirable. Conversely, there is solid evidence in the literature that anti-FLE antibodies can induce antibody-dependent enhancement of infection. The L107C/A313C stabilized sE dimer developed in this study does seem like a promising antigen as the authors claim because it faithfully presents EDE epitopes (unlike wild type sE) while preventing exposure of the FLE epitopes (unlike most virus particles). Moreover since the stabilized sE dimer lacks prM, it will not elicit anti-prM antibodies, which are abundantly expressed by dengue patients but show poor neutralization. Hence, is it plausible that as an antigen sE L107C/A313C will focus the immune response on production of the more effective anti-EDE antibodies and away from the less effective anti-FLE and anti-prM antibodies. The main concern is that there is no proof of principle that the stabilized dimer approach will work in dengue type 1 or that stabilized E dimers are indeed protective as antigens in animals (or humans).

Major concerns

1. One concern is that the authors were unable to produce dimeric sE from dengue type 1. As the authors are well aware, any dengue vaccine would have to target all four serotypes so inability to produce one of the four serotypes as a stabilized dimer could prove to be an

Achilles heel for this already technically demanding approach. Can a DENV2 sE L107C/A313C variant be produced with targeted mutations to reconstitute the DENV1 EDEs at the dimer interface?

2. There is no preliminary data to support that claim that stabilized E dimers are indeed protective as antigens in vivo. Although providing these data may fall outside the scope of the present study, the manuscript would be strengthened if the authors could anticipate some of the reasons why stabilized E dimers might not live up to expectations in animal model or clinical trials, and suggest possible workarounds.

Minor points

1. Lines 77-94. These two sections in the introduction are superfluous as they do not contain any information that is essential for the reader to understand the rest of the manuscript. Also, they break the flow between the previous and next paragraphs of the introduction.

2. Figure 6 is simple and small and belongs to the same Results subsection as Fig. 5. The two figures should be combined.

3. Lines 300-340. These four paragraphs read like a review of the literature and do not directly discuss any of the original data presented in this study. The authors should delete this section (and ideally use it in a separate review instead).

4. The electron density maps in Figure 4d-e looks as though they would benefit from B-factor sharpening for the purposes of model building and figure display (not refinement). If Bfactor sharpening produces more informative maps with greater side chain detail, the authors should include sharpened maps in figure 4 and list the sharpening factor applied. If sharpening helps the authors should also try improving the model, particularly of the double mutant structure, which currently has higher R-factors that would normally be considered acceptable.

Reviewer #1 (Remarks to the Author):

Rouvinski et al, attempts to produce a dengue E protein vaccine candidate that contains epitope that spans across the E proteins within a dimer and also at the same time, had their fusion loop hidden. This is because antibodies that bind across E protein (aka EDE antibodies) had been shown to be highly protective against the diseases, whereas those binding to fusion loop enhance the pathology of the disease. The authors generated different stable E protein dimers by making mutations on the E proteins leading to formation of disulfide bridge between the protomers. They characterized these mutants and showed that their structures are similar to the wild type E protein dimer, allowing EDE antibodies to bind. One of these mutants, the L107C/A313C, seemed to have successfully hid their fusion loops, therefore suggesting that this would be a strong vaccine candidate. The work is significant and important as it shows the possibility of making a construct that may elicit antibodies that bind quaternary structure dependent epitope, although the efficacy of this E protein dimer vaccine candidate is yet to be proven. The paper is overall very well written and easy to read.

Comments:

(1) Page 5 line 162: "C6/36 produced viruses have a much higher content of uncleaved prM compared to DENV produced in DC."

In other labs, some virus preparations from C6/36 can have low prM content, it could be a matter of prep to prep differences. Since the assumption of the high content of prM in your C6/36 is important for the interpretation of the ELISA competitive results, the authors should show a gel of the prM content of the virus that were used for these assays in the supplementary figures.

We have consistently observed high prM content in viruses produced in C6/36 cells and have previously compared the content to viruses produced in primary human dendritic cells using a ELISA to measure the prM and E content and used the ratio as a measure of relative prM content. Interestingly tumour cell lines such as vero and 293T also produce higher prM content viruses which can be reduced by overexpression of furin protease.

We have included a Western Blot of viruses produced in DC and C6/36 which shows a difference in prM expression (Fig S1) and modified the text of the paper (page 5 line 145).

(2) For figure 3a, the structures scored B, C D and D down the panel, respectively. Why is there is no structure that scored A?

The fact that none scored "A" means that the geometry observed in the crystal structure used for the prediction was not optimal at the corresponding sites for making a disulfide bond, and so if the protein were to be very rigid, no disulfide bond would have been formed. But as it is known that the E protein is rather flexible, there was the possibility that the polypeptide chain would adapt and disulfide bonds would form. We show in the paper that this is the case for two of the predicted

mutants, whereas for the other two this was not the case. The non-optimal geometry is also manifest in the fact that the two mutants that yielded covalent dimers, also yielded an important fraction of unfolded aggregates, as shown in Figs. 3b and S3a. The adaptation of the polypeptide chain to adopt a conformation compatible with disulfide bond formation is modest, especially for the double mutant, as shown by the RMSD of 0.14 Å for the double mutant and 1.11 Å for the single mutant (see new Fig. S5), which are values typically observed for the same protein crystallized in two crystal forms, for instance.

(3) Page 7, line 203, L017C should be L107C

Corrected, thank you for spotting this (page 6 line 188).

(4) Page 8, line 244, "The final refined structures of sE mutants have the same overall organization as wild-type sE...". Authors should provide a figure that superimposed the mutants with the WT E protein to show the similarities and also provide RMSD values.

We have prepared a new supplementary figure S5 with the superimposition of the DENV2-sE mutants with the wild type DENV2-sE in complex with Fab A11 (4UTB), together with the root mean square deviations for carbon alpha atoms.

(5) Page 9, last paragraph of the result section, authors suggests that A259C mutant has their fusion loop exposed and therefore able to insert into the liposome. However, the gel shows that most of the A259C are still at the bottom fraction and only some are at the top, unlike the WT DENV2 and DENV4 E proteins which showed more on the top fractions. Therefore, the paragraph should be toned down to:

"Some mutant A259C was still able to insert into liposomes and could be detected in the top fraction of the gradient (Fig. 6 lanes 3) whereas no protein co-floated with the liposomes in the L107C/A313C mutant (Fig. 6 lanes 5). These results suggest that in the A259C mutant the fusion loop is partially exposed and some of the proteins are able to interact with membranes in spite of the covalent linkage that prevents the sE protein from transitioning to the trimeric state and in line with the ability of FLE mAbs to bind A259C mutant in the ELISA assay (see above Fig. 5a)."

Thank you for the suggestion. We have used this sentence with some adaptations in the revised text (page 9 lines 275-282)

(6) Page 11 line 349, "all attenuated vaccines contain prM" – has this been determined? I would recommend deleted this unless there are publications or strong evidence on this.

What it was meant is that all live attenuated vaccines constructs carry prM, so inevitably there will be prM exposed. Whereas in the case of a subunit vaccine exposing just a stabilized sE dimer, there will be no prM exposed. The text was revised to make this clear (page 10 lines 315-317)

(7) Supplementary Figure 2b. DENV2 coomassie stained gel is not shown.

We apologize here, there was an error in the Figure S2 legend. The DENV2 Coomassie gel was actually displayed in Figure 3. We have removed the mention of DENV2 in the figure legend of Suppl. Fig. 3b.

Reviewer #2 (Remarks to the Author):

In this manuscript from Rouvinski and colleagues evaluate the possibility of using dengue virus engineered E protein dimers as immunogens in next generation vaccines. The authors compare the activities of two classes of neutralizing dengue antibodies: those that recognize the immunodominant fusion loop epitope (anti-FLE) and those that recognize the E dimer epitope anti-EDE). They elegantly show that anti-FLE bind to both monomeric and dimeric forms of E whereas anti-EDE bind to dimeric E and are potent and broadly neutralizing. They introduced cysteine substitutions that act to crosslink the E protein dimers and show that these are unaffected in structure and can be readily recognized by the two types of anti-EDE antibodies. Importantly, using a double crosslinked E dimer, but not a single, they show that the immunodominant FLE is no longer accessible. The authors argue that such a crosslinked dimer could be effectively used to direct the immune system toward a more potent, neutralizing antibody response, with a lower probability of eliciting ADE. The results presented are important and the experiments well executed. This manuscript represents an intellectual advance in the field and offers alternative considerations for next generation dengue virus vaccines.

Comments:

Figure 4: It is difficult to see the sites for cysteine substitution. Perhaps a blow up of this region in panels b and c would improve visualization of these sites.

The new version of Figure 4 now shows an enlargement of the structure in the regions of the two mutants, underneath each corresponding panel in 4b and 4c. The electron density maps corresponding to this region are shown in Figure S4.

Line 277: Although this floatation assay is provided to support their hypothesis that the single crosslink remains dynamic whereas the double substitution locks down the fusion loop exposure, can the authors provide any structural or biophysical evidence for this hypothesis? What about B factors from the two structures?

Thank you for this comment. Unfortunately, at 3.9Å resolution there are not enough observations to accurately refine the B factors. It is important, however, to mention

that the structures correspond to a complex of the sE mutants with an EDE2 antibody, which binds over the fusion loop and stabilizes it. We nevertheless did not observe higher B factors on the fusion loop region, as expected because their mobility is constrained by the bound Fab. The fact that the FLE antibodies still efficiently bind the single mutant, and that this mutant can still partially float with liposomes on a sucrose gradient at low pH, provide a strong argument to postulate that they are exposed through a breathing motion about the central disulfide bond, as proposed in the text

Discussion: The Discussion on page 10 reads like a review and should be truncated.

Thank you, this has been done.

Supplementary Table 1: The PDB coordinates need to be added.

The PDB code are now added now (page 23 line 764)

Reviewer #3 (Remarks to the Author):

In this study Rouvinski et al describe the structure and antibody binding properties of variants of the dengue virus envelope glycoprotein, E, which were engineered to be constitutively dimeric through the introduction of cysteine residues that form intermolecular disulfide bonds across the E dimer interface. The authors show that most antibodies against the immunodominant fusion loop epitope (FLE)- 36 out of 46 tested- fail to bind a soluble E fragment with two engineered intermolecular disulfides (sE L107C/A313C) in an ELISA. In contrast, most antibodies against the E dimer epitope (EDE)- 28 out of 42 tested- were able to bind the sE L107C/A313C variant. This stands in sharp contrast to the wild type sE protein, which binds all 46 anti-FLE antibodies and only one or two of the EDE antibodies. This is significant because anti-EDE antibodies were shown by a similar group of authors to be broadly neutralizing against all four dengue serotypes, whereas anti-FLE antibodies can be potentially harmful as they are often poorly neutralizing, potentially induce antibody-dependent enhancement of infection, and are the most type of antibody produced most abundantly during dengue virus infection. Hence the authors claim that stabilized E dimers such as sE L107C/A313C could serve as a broadly protecting subunit vaccine against dengue virus, and also against Zika virus, in which many of the EDEs are conserved.

These key findings and central claim are underpinned by various supporting data including: (1) biochemical studies showing that anti-EDE antibodies stabilize wt sE and that the sE L107C/A313C variant is a dimer in solution, (2) crystal structures of sE L107C/A313C and another stabilized sE variant in complex with an EDE antibody showing that the EDE epitopes are intact in the sE variants, (3) a competition ELISA showing that dengue virus particles produced mosquito cell (which have more exposed FLE epitopes than viruses produced in human cells) preferentially bind to anti-FLE antibodies rather than anti-EDE antibodies when both are present, and (4) sE L107C/A313C variants from dengue serotypes 2, 3

and 4 did not interact with liposomes, whereas the wild type and single-disulfide variants did, confirming that the fusion loop (and FLE) is not exposed in the sE L107C/A313C.

This manuscript is well written, timely and overall quite convincing. The authors showed previously that anti-EDE antibodies broadly and potently neutralize dengue viruses from all four serotypes. Eliciting this type of antibody during infection is therefore desirable. Conversely, there is solid evidence in the literature that anti-FLE antibodies can induce antibody-dependent enhancement of infection. The L107C/A313C stabilized sE dimer developed in this study does seem like a promising antigen as the authors claim because it faithfully presents EDE epitopes (unlike wild type sE) while preventing exposure of the FLE epitopes (unlike most virus particles). Moreover since the stabilized sE dimer lacks prM, it will not elicit anti-prM antibodies, which are abundantly expressed by dengue patients but show poor neutralization. Hence, is it plausible that as an antigen sE L107C/A313C will focus the immune response on production of the more effective anti-EDE antibodies and away from the less effective anti-FLE and anti-prM antibodies. The main concern is that there is no proof of principle that the stabilized dimer approach will work in dengue type 1 or that stabilized E dimers are indeed protective as antigens in animals (or humans).

Major concerns

1. One concern is that the authors were unable to produce dimeric sE from dengue type 1. As the authors are well aware, any dengue vaccine would have to target all four serotypes so inability to produce one of the four serotypes as a stabilized dimer could prove to be an Achilles heel for this already technically demanding approach. Can a DENV2 sE L107C/A313C variant be produced with targeted mutations to reconstitute the DENV1 EDEs at the dimer interface?

We thank the reviewer for raising this issue. We have made the following addition to the discussion:

We will attempt to express DENV1 dimers in alternative expression systems, however, as the EDE is a cross-reactive epitope, the EDE MAbs efficiently neutralize DENV1 as well as the other serotypes. In the future, priming and boosting by alternating the three DENV serotypes for which the dimers can be obtained should be sufficient to preferentially stimulate B cells producing EDE antibodies, which by definition would also neutralize DENV1. The possibility remains to resurface sE from DENV3 (which is most closely related to DENV1) in the event that a ENV1-like sE molecule with a stable EDE is required, a similar approach has recently been used to graft the DENV3 specific epitope for mAb 5J7 into DENV1²⁸. (page 9 Lines 292-299)

2. There is no preliminary data to support that claim that stabilized E dimers are indeed protective as antigens in vivo. Although providing these data may fall outside the scope of the present study, the manuscript would be strengthened if the authors could anticipate some of the reasons why stabilized E dimers might not live up to expectations in animal model or clinical trials, and suggest possible workarounds.

We have added the following to the revised discussion:

“Testing the ability of DENV stabilized dimers to elicit an anti-EDE response may be complicated as murine antibody responses differ considerably from human responses in particular murine responses are much more directed to domain III of E and complex conformational epitopes appear rare. Furthermore, we anticipate focussing the response to the EDE may require heterologous prime boosting strategies.” (page 10 lines 331-335)

Minor points

1. Lines 77-94. These two sections in the introduction are superfluous as they do not contain any information that is essential for the reader to understand the rest of the manuscript. Also, they break the flow between the previous and next paragraphs of the introduction.

We have removed these sections

2. Figure 6 is simple and small and belongs to the same Results subsection as Fig. 5. The two figures should be combined.

Thank you for this suggestion. Both Figures were combined now in the current Figure 5.

3. Lines 300-340. These four paragraphs read like a review of the literature and do not directly discuss any of the original data presented in this study. The authors should delete this section (and ideally use it in a separate review instead).

This has been deleted

4. The electron density maps in Figure 4d-e looks as though they would benefit from B-factor sharpening for the purposes of model building and figure display (not refinement). If B factor sharpening produces more informative maps with greater side chain detail, the authors should include sharpened maps in figure 4 and list the sharpening factor applied. If sharpening helps the authors should also try improving the model, particularly of the double mutant structure, which currently has higher R-factors that would normally be considered acceptable.

Thank you for this suggestion. Fig. S4 now displays the electron density with and without B-sharpening. We have found that in certain regions of the map, B-sharpening indeed gave better interpretable densities. Model building using the sharpened B factors did help obtain a better R factor for the single mutant. For the double mutant, the diffraction was much weaker and the quality of resulting data set was substantially poorer, and we were not able to improve the R factors in that case.

REVIEWERS' COMMENTS:

Reviewer #1 (Remarks to the Author):

I am happy with the changes.

Reviewer #2 (Remarks to the Author):

The authors have adequately addressed the comments of the reviewers. The manuscript is substantially improved as a result of the authors' changes. Therefore, this manuscript will contribute in a significant way to understanding the antibody response to dengue viruses and is consistent with publication in Nature Communications.

Reviewer #3 (Remarks to the Author):

My concerns have all been adequately addressed in the current revision.